# Detection and Quantification of Cracking in Concrete Aggregate through Virtual Data Fusion of X-Ray Computed Tomography Images

**DOI:** 10.3390/ma13183921

**Published:** 2020-09-04

**Authors:** Tyler Oesch, Frank Weise, Giovanni Bruno

**Affiliations:** 1Bundesanstalt für Materialforschung und–prüfung, BAM (Federal Institute for Materials Research and Testing), 12205 Berlin, Germany; frank.weise@bam.de (F.W.); giovanni.bruno@bam.de (G.B.); 2Institute of Physics and Astronomy, University of Potsdam, Karl-Liebknecht-Str.24-25, 14476 Potsdam, Germany

**Keywords:** X-ray computed tomography (CT), concrete, alkali-silica reaction (ASR), ASR-sensitive aggregate, solubility test, specific surface area, crack detection, automated image processing, damage quantification

## Abstract

In this work, which is part of a larger research program, a framework called “virtual data fusion” was developed to provide an automated and consistent crack detection method that allows for the cross-comparison of results from large quantities of X-ray computed tomography (CT) data. A partial implementation of this method in a custom program was developed for use in research focused on crack quantification in alkali-silica reaction (ASR)-sensitive concrete aggregates. During the CT image processing, a series of image analyses tailored for detecting specific, individual crack-like characteristics were completed. The results of these analyses were then “fused” in order to identify crack-like objects within the images with much higher accuracy than that yielded by any individual image analysis procedure. The results of this strategy demonstrated the success of the program in effectively identifying crack-like structures and quantifying characteristics, such as surface area and volume. The results demonstrated that the source of aggregate has a very significant impact on the amount of internal cracking, even when the mineralogical characteristics remain very similar. River gravels, for instance, were found to contain significantly higher levels of internal cracking than quarried stone aggregates of the same mineralogical type.

## 1. Introduction

### 1.1. Alkali-Silica Reaction (ASR)

Despite decades of research, the problem of harmful alkali-silica reaction (ASR) in the field of concrete construction has not yet been satisfactorily solved. For the first time in 1940, Stanton [1] reported damaging strains within concrete due to chemical reactions of cement and aggregate. In the 1950s, Powers and Steinour [2,3] developed initial models of ASR’s damage mechanism. In the 1970s, Locher and Sprung [4] identified opal and porous flint as alkali-sensitive aggregates and developed theories on their reaction mechanisms. In the 1980s, various researchers conducted in-depth studies on the influence of alkali metal salts on the swelling pressures of the ASR gel [5,6]. The current state of knowledge in the field of ASR has also been extensively described in number of recent publications [7,8].

During the ASR process, the reactive SiO_2_ within aggregates reacts with alkalis (supplied from the cement paste or from an external source) in the presence of water to form expansive alkali silicate hydrates (ASR gels). Because the tensile strength of road surface concrete is often significantly lower than the swelling pressures caused by ASR gels, cracking can be induced [9]. The progression and the extent of the resulting cracking processes are determined to a large extent by the type of aggregate. For example, fast-reacting aggregates (among others, flint, opaline sandstone and mudstone) are characterized by gel and crack formation emanating from the transition zone between the grain and the mortar matrix [7]. On the other hand, in the case of the slow-reacting aggregates (for instance greywacke or quartz porphyry), which were of primary of interest in this project, the gel formation takes place above all inside the aggregate itself, which results in the formation of internal aggregate cracks [7].

The severity of the ASR degradation process is thought to be partially dependent on the amount of porosity within a given aggregate that is accessible to liquid penetrating from the sample surface. Against this background and to evaluate the alkali sensitivity of the aggregate, the influence of the specific surface area on the solubility behaviour of four different aggregates in 0.1 M potassium hydroxide solution without and with defined addition of NaCl at a temperature of 80 °C was thoroughly investigated in a joint project [10]. To quantify this relationship, a non-destructive method is needed for measuring both the external surface area of aggregates and the internal surface area of aggregate cracks and pores, including a differentiation of internal voids connected to the sample surface from those isolated from the surface. The primary focus of this publication is devoted to the crack detection method. A detailed analysis of the implications for ASR damage, including a comparison of the CT results with those from other porosity measurement methods, such as mercury porosimetry and the Brunauer–Emmett–Teller (BET) method, can be found in Oesch et al. 2020 [11] and in Weise et al. 2019 [10].

### 1.2. X-Ray Computed Tomography (CT)

The development of X-ray computed tomography (CT) began in the 1960s and clinical X-ray CT investigations have been widely conducted since the 1970s [12,13]. Since that time, many different reconstruction algorithms have been developed for clinical use, including algorithms based on the algebraic reconstruction technique (ART), filtered back projection (FBP), and iterative reconstruction (IR) [14]. Medical X-ray CT scanning systems are, however, unsuitable for many applications in materials science given their lower X-ray energy characteristics and coarser resolution compared to specialized laboratory-based X-ray CT systems used for materials research [15]. These differences occur both because the size of the intended scanning objects tends to significantly differ between clinical and materials science applications and because the X-ray absorption characteristics of live tissues are much lower than those of materials such as concrete and steel.

X-ray CT has been used in non-destructive concrete research applications for more than 30 years [16,17]. In this scanning method, a sample is placed on a rotating table between an X-ray source and an X-ray detector [18]. By adjusting the distances between the X-ray source, the sample and the X-ray detector, it is possible to vary the voxel (i.e., 3D pixel) resolution in the resulting images. The penetration of the sample by the X-ray beam causes an X-ray attenuation image of the sample to be projected upon the detector. By recording these projected images during the 360° rotation of the sample, the projections can be inverted using volume reconstruction algorithms, which produce a 3D representation of X-ray attenuation within the sample [19]. X-ray attenuation is approximately proportional to local material density and can be used to identify single objects within a material (or structure) and to individually separate and analyze those objects.

Previous research has shown that X-ray CT scans can be taken during incremental testing. This includes mechanical testing (such as unconfined compression, split cylinder, triaxial, and reinforcing bar pull-out testing [20,21,22,23,24]), chemical testing (such as the measurement of progressive corrosion in reinforced concrete during repeated exposure to chloride [25] and the transport of water [26,27,28]) and thermal testing (such as water migration in heated concrete [29]).

### 1.3. Crack Detection and Quantification

Crack detection and quantification is important for understanding and modelling a series of material behaviours. Precise measurement of crack surface area is needed, for instance, in order to calculate the fracture energy expended during damage processes using basic fracture mechanics relationships [30]. The crack orientations have also been observed to exhibit a behaviour that is highly dependent on the anisotropy of the material structure, such as fibre orientation within fibre-reinforced concretes [31].

Crack detection and quantification within X-ray CT images have been the subject of extensive past research. Most of these crack detection methods have leveraged one or more unique characteristics of cracks, which differentiate them from the surrounding material. Possibly the most popular research approaches have focused on the use of template-matching methods in order to separate cracks from the surrounding materials [32,33]. This method of crack detection relies on the similarity of cracking structures to certain template shapes, such as small planes or discs. Although impressive results have been demonstrated using the template-matching method, the template parameters are not universal and must generally be tailored for each material and each imaging scenario.

Research was recently carried out by Paetsch (2019) [34] with the goal of partially overcoming these challenges related to using template matching approaches. This research indicated that the results of a series of analyses carried out using different template shapes can be combined in order to obtain a greater accuracy of the detected cracks. However, Paetsch (2019) [34] has underlined that further problems remain to be solved that are common to most template-matching methods, such as difficulties detecting cracks in areas where significant crack branching or widening occurs.

Another method that takes advantage of the narrow shape characteristics of cracks is a Hessian-based approach [35,36], which identifies regions that exhibit sharp changes in image intensity. Percolation methods have also shown significant promise in detecting cracks across a range of materials [37]. These methods leverage the fact that most cracks are continuous, narrow objects with relatively consistent (low) density. Percolation methods have proven insufficient, however, to accurately detect complex cracks of varying size in most materials. Impressive results have also been obtained through a combination of the Hessian and percolation-based methods into the Hessian-driven percolation approach, although the processing time required for such an analysis remains prohibitive for most high-resolution CT images [32].

Many methods have also been employed that leverage the unique characteristics of a specific material or damage scenario. One excellent example of this approach is the use of digital volume correlation (DVC) to detect cracks in samples subjected to in-situ loading [38,39,40]. DVC is used to measure strains within samples by calculating voxel movements between subsequent CT images (such as images calculated before and after a loading increment). Cracks can, thus, be identified as areas of either high strain or poor correlation within DVC images or through more complex analysis methods, such as phase-congruency analysis. Although these methods appear to exhibit a relatively high accuracy (even sub-pixel, see [41]), they are only useful for detecting cracks caused by progressive in-situ testing with simultaneous CT. They do not provide any benefit for detecting cracks that are already present in specimens prior to testing. Phase congruency can, however, also be used to detect edge features (including cracks) within the greyscale images without direct reference to DVC [42].

Another example of a material-specific crack detection approach is the leveraging of typical wood structure within logs to identify cracks. These cracks typically run perpendicular to the growth rings of the logs and can, thus, be easily identified by their orientation characteristics [43].

Despite the many promising crack detection methods outlined here, a series of obstacles remain that have prevented the implementation of consistent, accurate, and quantitative crack analyses as part of CT scanning. First, most of these methods require some amount of tailoring for specific material properties (as in the case of template matching and percolation) or specific crack conditions (such as measuring only those arising from in-situ testing using DVC methods). Second, none of these methods has been successfully validated for the quantitative determination of crack properties (such as surface area) based on other standard measurement techniques.

In this study, we will show, on the example of different aggregate types, how the obstacles mentioned above can be circumvented by a novel data fusion strategy.

## 2. Materials and Methods

### 2.1. Sample Selection

For this testing series, a group of aggregates from four different categories were selected and analysed. These categories were selected in order to include stones with a variety of different mineralogical compositions, deterioration conditions, and alkali-sensitivity characteristics (Table 1 and Figure 1) [44]. All stones included in this analysis were in the 8 mm to 16 mm size range: During the sieving process, the individual stones passed successfully through a sieve with a 16 mm mesh but were unable to pass through a sieve with an 8 mm mesh [45].

For the categories GK1 and GK4, the selected stones were sieved from a natural river gravel. Such river gravels are typically characterized by significant mineralogical deterioration due to naturally occurring weathering processes. In order to characterize these river gravels, which have a heterogeneous mineralogical composition, the primary types of rock (7 types in total) occurring in GK1 and GK4 were determined and 10 individual grains of each type were selected for CT analysis.

For the categories GK2 and GK3, the selected stones were sieved from crushed stone chips that were quarried from solid rock deposits. Such quarried stones are typically characterized by greater mineralogical integrity than river gravels because they have not been exposed to significant weathering. Due to the homogeneity of the quarried stone, the selection of individual grains was limited to 10 each for GK2 and GK3. Thus, combining all the individual grains from each of the four stone categories included in this research study, a total of 90 different individual grains were investigated.

### 2.2. CT Scanning

During this research program, an acceleration voltage of 130 kV and current of 180 µA were used for the X-ray source. The X-ray beam was also filtered using a 0.5 mm thick Copper plate immediately upon leaving the source in order to remove (unwanted) photons of small energies from the X-ray beam, thereby increasing the contrast of the resulting images. The flat panel detector used for this scanning contained a 2048 × 2048 pixel field.

Individual aggregates were sorted based on mineralogical characteristics and placed within corresponding plastic tubes with small pieces of foam separating the aggregates from one another. As a result of heating and deterioration of the target material within the X-ray tube as well as changes in detector sensitivity over time, significant variations in the measured X-ray beam intensity and distribution can occur. To compensate for these variations, “dark-field” and “bright-field” images, which correspond to blank images (i.e., containing no sample) acquired with no illumination and full illumination, respectively, were acquired prior to the scanning of each plastic tube of samples. These images were then used to calibrate the X-ray images of the samples.

The CT machine was pre-programmed to collect a complete scan of each aggregate before repositioning the plastic tube using a manipulator and beginning the scan of the next aggregate. Thus, the scanning conditions for all stones within any given plastic tube were identical. All scanning conditions other than resolution were also held constant for all plastic tubes.

It was important to maintain scanning conditions that were as consistent as possible to ensure that the results of the crack analysis would be comparable. In spite of this, the resolution was maximised for each stone type, if they significantly varied in size and shape. This was done by adjusting the distance between the plastic tube and the X-ray source. The corresponding voxel sizes were then calculated directly from the measured distances between the X-ray source, the sample holder and the X-ray detector for each individual set of scans.

Although these variations in image resolution are known to directly affect measurements of crack properties, such as surface area (increased surface area is generally detected with decreasing voxel sizes), some estimation of the magnitude of this effect can already be accounted for based on a recent study [46]. The results of this study indicate that even a doubling of voxel size does not appear to generally change the measured crack surface area by more than a factor around two. We will see below that the small voxel size variations had little influence on the much larger variations in measured crack surface area among the different aggregate types, so that comparisons could be made, and conclusions drawn.

### 2.3. Image Analysis

#### 2.3.1. Data Fusion Approach

In such an environment, where many analysis methods are readily available, but no individual method is sufficiently accurate to provide the needed measurements, the use of a data-fusion inspired approach becomes very promising and attractive. Data fusion enables researchers to combine data from multiple sources in order to produce more consistent and accurate results than those provided by any single source [47]. In non-destructive testing, the different sources used in data fusion typically result from differing non-destructive measurement approaches. Such data-fusion based approaches have even been successfully applied to the application of crack detection [48].

It is, however, often the case that researchers only possess meaningful data from a single measurement technique, such as CT (often indeed used as a reference), or do not have access to further non-destructive testing equipment. Even in this case, it should be possible to use the theory behind the data fusion approach to improve the overall quality of the quantitative determinations resulting from image analysis. Rather than relying on a wide variety of physical measurement techniques, data fusion in this approach would be carried out by using the output from an array of different image analysis techniques. Since such an approach relies on the fusion of different computationally generated data sources resulting from the same original CT image rather than on a variety of different physical measurement techniques, it can be more appropriately described as virtual data fusion. This is similar to the process that is thought to occur when an expert identifies cracks using the human eye. The expert has only one measurement technique (a visual image), but by considering its many different aspects (such as coloration, shape, and relationship to surrounding objects or planes of stress), the expert is able to quite easily and accurately detect a crack on the surface of a specimen.

Figure 2 shows a diagram of this virtual data fusion approach. In the diagram, the characteristics of a crack are displayed in dark grey boxes and the image processing steps for identifying objects with those characteristics are depicted in light grey boxes. The list provided here is only meant to serve as an example and is by no means exhaustive. It is clear, however, that results obtained through a fusion of the results from twelve such independent analyses will be much more accurate and resilient to varying material conditions than the results from any single analysis technique. An example of how such a step-by-step approach can identify individual cracks within a generic concrete sample is shown in Figure 3.

#### 2.3.2. Implementation

Although, in theory, twelve or more different analysis methods could be used in the virtual data fusion implementation for analysing the ASR-sensitive aggregate dataset, in practice such a full implementation was impractical. We selected a partial implementation of the scheme including only three analysis approaches. The choice included the approaches that would produce cracking data with sufficient accuracy for this specific application while simultaneously testing the effectiveness of the virtual data fusion concept for obtaining accurate and consistent results. A successful demonstration of the virtual data fusion method with only three analysis components would then give justification for the further development of the algorithm to gradually include additional analysis modules, simultaneously growing in accuracy and resiliency. Moreover, the modular architecture of the strategy could more easily be adapted to different problems than a full but rigid analysis.

In fact, for the specific ASR-sensitive aggregate analysis described in this paper, it was not important to separate internal pores from internal cracks since the surface area of both pores and cracks was vulnerable to ASR degradation. Thus, modules related to this differentiation could be left out of the analysis. Furthermore, since scans were not available at varying levels of degradation, all methods relying on changes in sample state relative to time (such as DVC-based methods) were also left out of this analysis. All image analysis described in this paper was completed using custom algorithms developed and implemented in MATLAB [49].

##### Module 1: Identification of Objects with Low Density

When cracks or pores have widths exceeding two voxels, the voxels in their centres are completely filled with air. Thus, these central voxels are characterized by a particularly low greyscale value. In order to separate these voxels, a threshold has to be selected and subsequently used for image binarization. All voxels darker than the threshold would then be transformed to white and all voxels lighter than the threshold transformed to black. Given that this threshold must be consistently applied for a large range of datasets, an automated selection method was implemented.

Using the triangle selection algorithm [50,51], a virtual line is drawn from the origin of the image histogram to the top of the largest histogram peak (excluding the initial peak at zero, which represents voxels from the air around the specimen) (Figure 4). A calculation is then conducted to determine which point on the histogram is furthest from the virtual line along an intersecting, perpendicular line. The location of that point is identified as a potential threshold. It was found, however, that such a threshold is rather over-encompassing, leading to the introduction of considerable noise into the resulting binarized images. Thus, a slightly more conservative threshold value was also implemented for this module that is 25% lower than the original threshold. An example of the results from this analysis can be observed in Figure 5.

##### Module 2: Identification of Objects with High Gradient

When an object in a CT-image has either a much higher or a much lower greyscale value compared to the surrounding material, its edges can be identified as regions of high gradient. For this purpose, a gradient-magnitude image is calculated from the original CT-image through the use of the Sobel operator (Figure 6) [52]. Since the resulting gradient image consists of a wide range of grey values, it must also be binarized. For this purpose, a histogram of the gradient image grey values is computed and another automated triangular threshold selection is completed, this time from the right edge of the histogram (Figure 7). The resulting binary image contains the edges of both bright and dark objects in the original CT image (Figure 8). In order to remove the edges associated with the bright (high-density) objects in the CT-image as well as some of the noise, the binarized gradient image can be multiplied by a binarized CT-image (this time using the “over-encompassing void-solid threshold” identified in Figure 4) (Figure 9).

##### Module 3: Identification of Objects above a Given Size

Prior to noise identification and removal, the results from Modules 1 and 2 were combined into a single image (Figure 10). This ensured that the centres and edges of the cracks/pores would both be present and accounted for during object size analysis. The object size analysis was completed using a connected components algorithm. During this analysis, individual voxels are assessed to determine whether they are part of a larger object in the binary image by analysing whether they adjoin other voxels of the same color. For this analysis, white voxels with touching faces, edges or corners (also known as “26-connected”) were identified as connected. Objects with voxel volumes smaller than 125 voxels were subsequently identified and eliminated (Figure 11).

It was found that the use of smaller cubes than 125 voxels (with five voxel long sides) tended to leave considerable noise within the image while the use of larger cubes resulted in the loss of a considerable number of voxels along the path of the cracks. Given that the full CT images each contained over seven billion voxels, such an object with a volume of 125 voxels represented less than one ten-millionth of the total image volume. A flowchart of the entire crack-detection process is provided in Figure 12.

## 3. Results

After the cracks and internal voids were identified within each of the aggregates, their characteristics could be quantified and compared. Of primary interest for this analysis was the determination of volume and surface area characteristics. In particular, in order to compare the results of the CT-analysis with those of other non-destructive measurement techniques, it was important to separate surface-connected cracks and voids (referred here to as “open voids”) from internally isolated cracks and voids (referred to here as “closed voids”). This is because measurements of internal surface area using the Brunauer–Emmett–Teller (BET) method [53,54] only account for internal voids accessible from outside of the sample. Similarly, the measurement of void volume through mercury porosimetry is thought to depend primarily on the saturation of internal voids that are connected to the stone surface. Such a separation of surface-connected cracks could be completed using a connected components analysis in which only cracks and voids containing voxels that touched the stone surface were retained.

The crack/pore surface area measurements obtained using this approach are provided for all of the river-gravel type aggregates in Figure 13 and for both river-gravel and quarried-stone aggregates of the minerals rhyolite and greywacke in Figure 14. Note that the measurements for the greywacke (GK4) and rhyolite (GK1) river gravels appear in both figures for comparison purposes. Figure 15 also provides 3D images of cracking distributions for two selected individual grains of the same mineral (greywacke), where one grain has been extracted from a quarry and the other has been taken from river gravel. For images of all analysed aggregate types, see Appendix A. Tabulated values of the surface area measurements are also provided for each individual sample in Appendix B.

Error bars in Figure 13 and Figure 14 could not be calculated. This is because there is still no universally accepted method for estimating the combined error introduced by CT measurement systems and image processing algorithms. Although numerous approaches for estimating such error bars have been proposed [55,56], these tend to be rather computationally intensive and time consuming and remain an active area of research. For pure dimensional measurements, it is common to use either the voxel size or the focal spot size of the X-ray tube as an estimation of error. Given that the focal spot size of this scanning system was significantly smaller than the voxel sizes obtained during these investigations, the voxel sizes for each scan (also listed in Figure 13 and Figure 14) can be taken as an estimation of possible error in the dimensional measurements.

From Figure 14 it is clear that the amount of internal cracking (including both surface-connected cracking and non-surface-connected cracking) in the river gravel aggregates was much higher than that in the quarried aggregates, even when their mineralogical characteristics were similar. The magnitude of this effect is also much too large to be attributed to variations in CT resolution. The underlying basis for these differences in quantitative crack measurements can also be estimated through visual observation of CT images, such as those displayed in Figure 15. Clear, layered cracking is visible within river gravel greywacke stones; this is not present within the greywacke stones extracted from quarries. This is thought to result from the aggressive weathering process that river gravel is subjected to during its lifecycle prior to construction use. This indicates that the selection of high-quality aggregate based on mineralogical characteristics alone may be insufficient.

It is also clear from Figure 13 that even for stones from a single source and with a single mineral composition, a significant amount of variability in internal porosity and cracking is present. The variation between individual stones of a single type (such as rhyolite (GK1)) is often much larger than the average difference between two entirely different stone types (such as between rhyolite (GK1) and granite (GK4)). Thus, we recommend the use of large statistical samples to properly characterize each stone type for ASR sensitivity.

## 4. Discussion and Conclusions

This research clearly demonstrates the need for universal, automated, and consistent crack detection methods that allow the cross comparison of results from large quantities of CT-scan data from different sample types. A framework, called “virtual data fusion“, was developed that has the potential to successfully provide such a method. A partial implementation of this method in a custom program was developed for use in research focused on crack measurement in ASR-sensitive aggregates. Our results demonstrated the success of the program in effectively identifying crack-like structures and measuring their characteristics such as crack extension (relative surface area) and surface connectivity.

These results demonstrate the significant impact that the source of extraction can have on the characteristics of aggregates. Even for aggregates of the same mineral type, river gravels contain significantly higher levels of internal porosity and cracking than quarried stone. This is thought to result from the aggressive weathering process that river gravel is subjected to prior to its selection and use for construction. This indicates that the selection of high-quality aggregate based on mineralogical characteristics alone may be insufficient. It is also clear from these results that there is a significant amount of variability in internal porosity and cracking even for stones with the same mineralogical characteristics and extraction source. Thus, large statistical samples will be necessary to properly characterize each stone type for ASR sensitivity.

## Figures and Tables

**Figure 1 materials-13-03921-f001:**
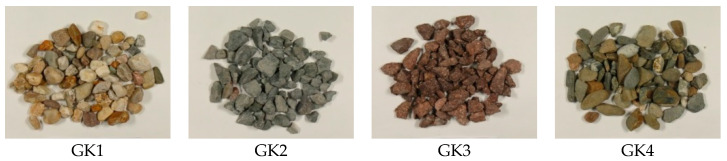
Photographs of typical individual grains from each of the stone categories.

**Figure 2 materials-13-03921-f002:**
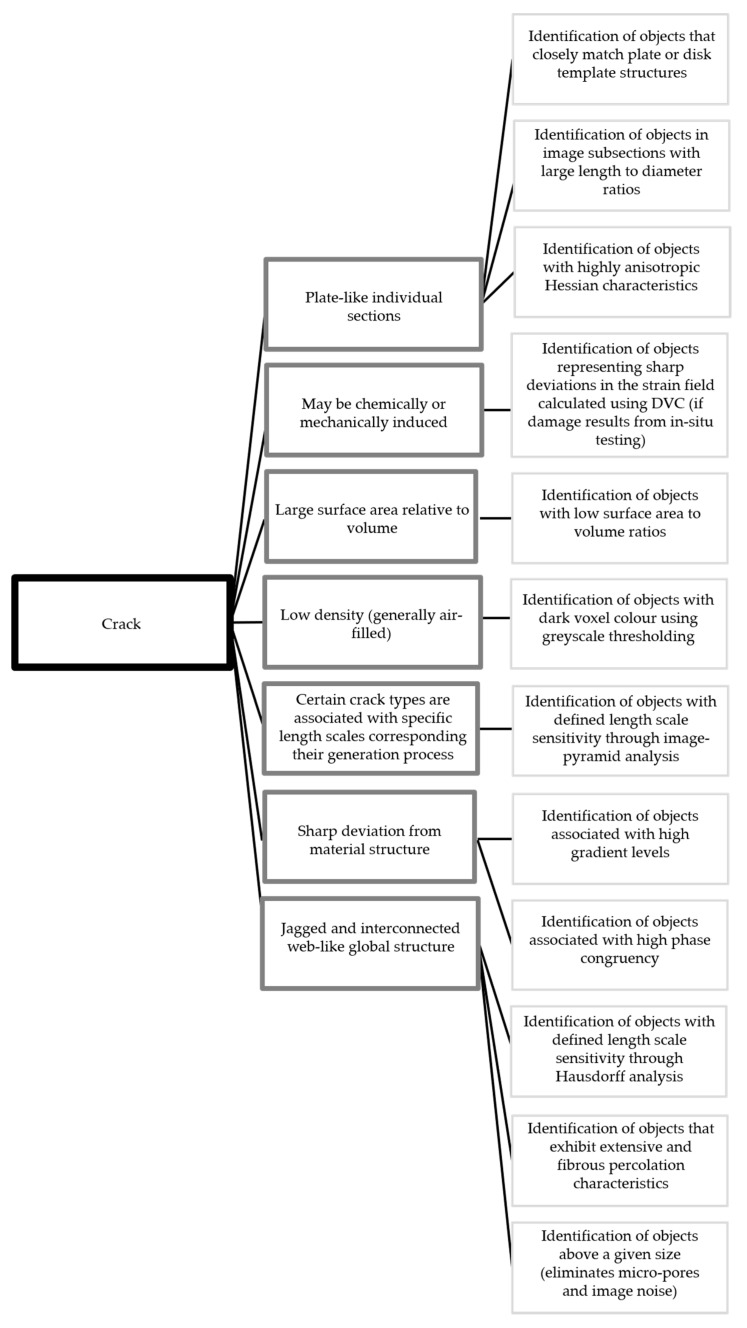
Virtual data fusion web linking cracks (black) with their characteristics (dark grey) and image analysis techniques for identifying objects with those characteristics (light grey).

**Figure 3 materials-13-03921-f003:**
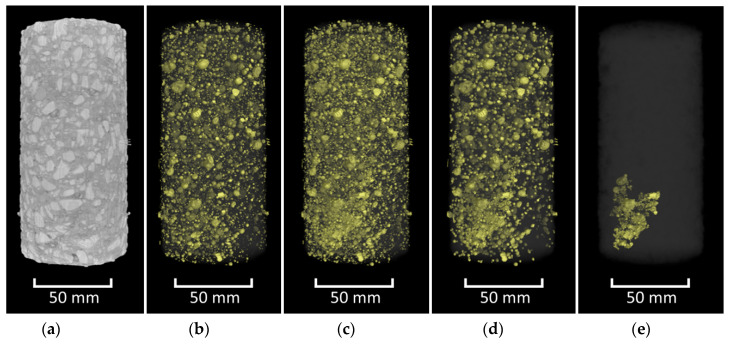
Step-by-step isolation of cracking by means of virtual data fusion. CT image of a concrete sample (**a**), voids identified using a greyscale threshold (**b**), voids identified in (**b**) added to interfacial zones identified using a gradient-based analysis (**c**), removal of small connected components (noise and isolated pores) from image (**d**), and elimination of components with small specific surface areas from image (**e**). Raw data courtesy of U.S. Army Engineer Research and Development Center (ERDC).

**Figure 4 materials-13-03921-f004:**
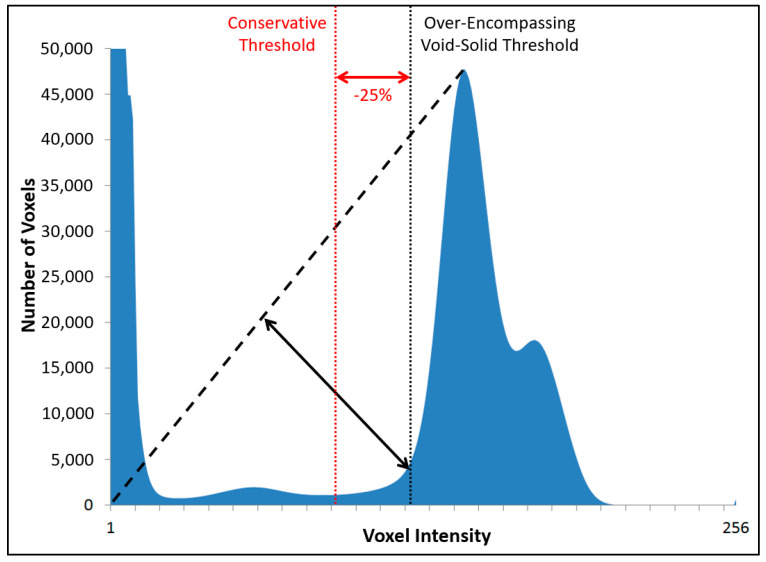
Automated threshold selection using the triangle approach (based on figure in [10]).

**Figure 5 materials-13-03921-f005:**
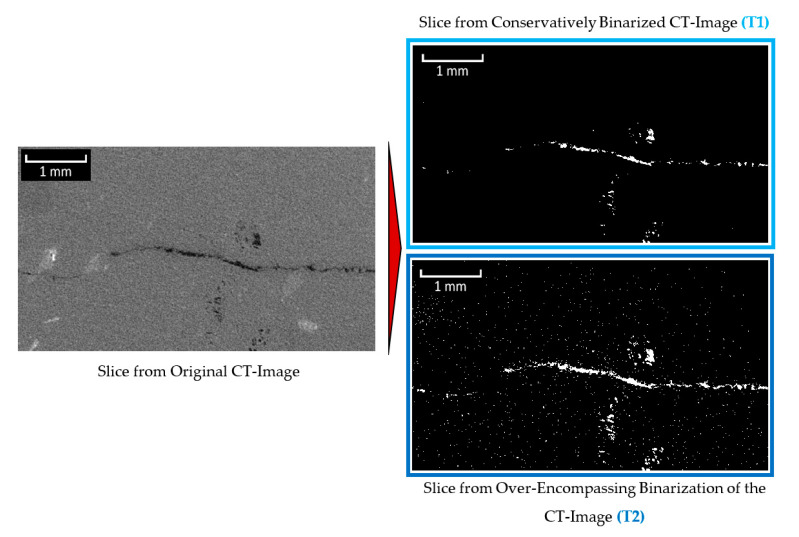
Binarization of greyscale image using the conservative and over-encompassing thresholds identified using the automated selection method (based on figure in [10]).

**Figure 6 materials-13-03921-f006:**
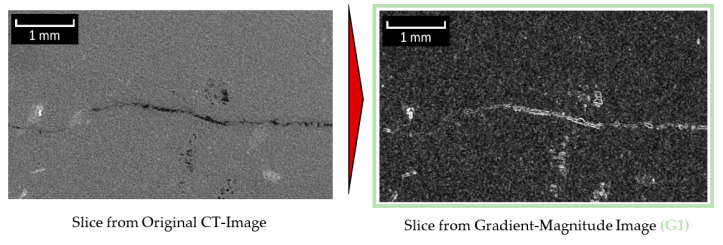
Generation of the gradient-magnitude image (based on figure in [10]).

**Figure 7 materials-13-03921-f007:**
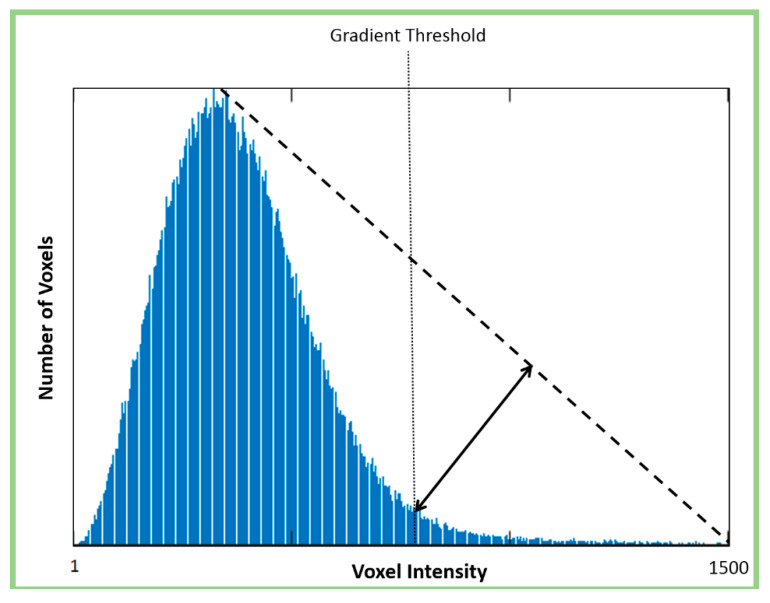
Automated gradient threshold selection using the triangle approach (based on figure in [10]).

**Figure 8 materials-13-03921-f008:**
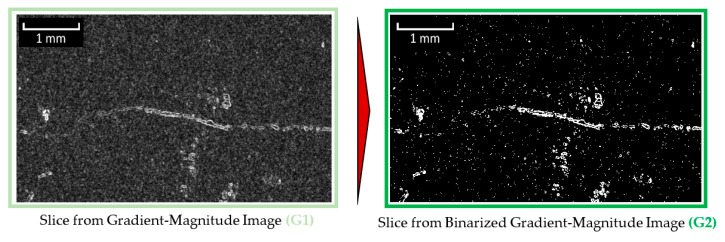
Binarization of the gradient image (based on figure in [10]).

**Figure 9 materials-13-03921-f009:**
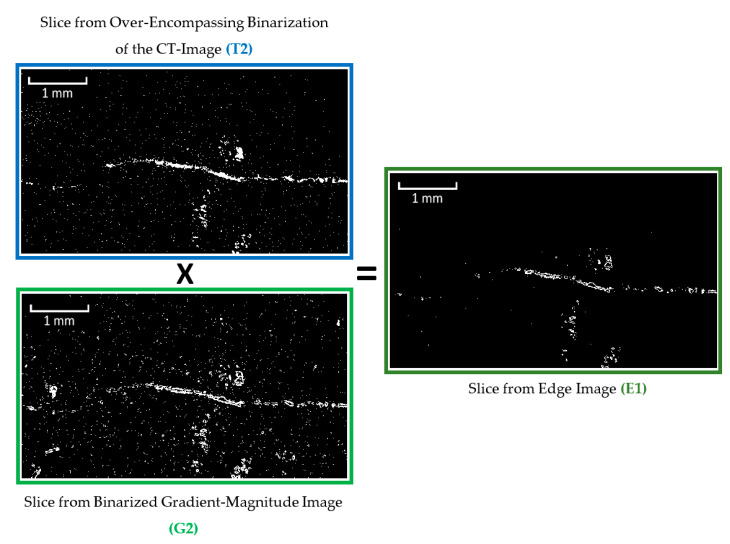
Generation of an image containing the outer edges of only low-density objects (based on figure in [10]).

**Figure 10 materials-13-03921-f010:**
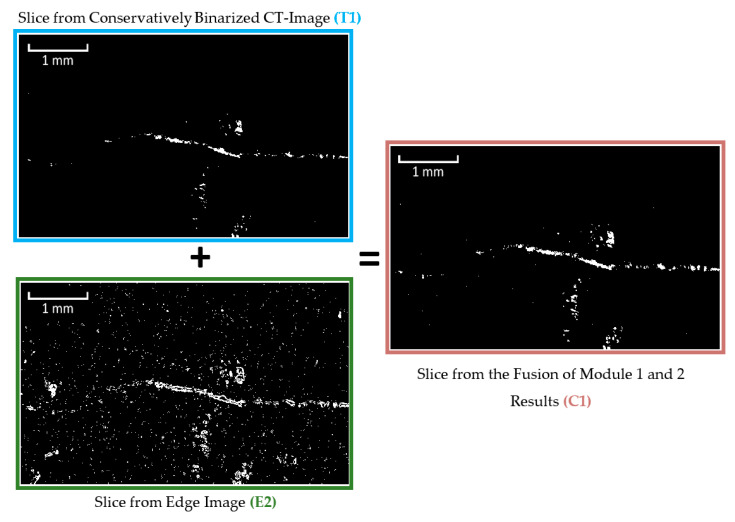
Fusion of images resulting from Modules 1 and 2 (based on figure in [10]).

**Figure 11 materials-13-03921-f011:**
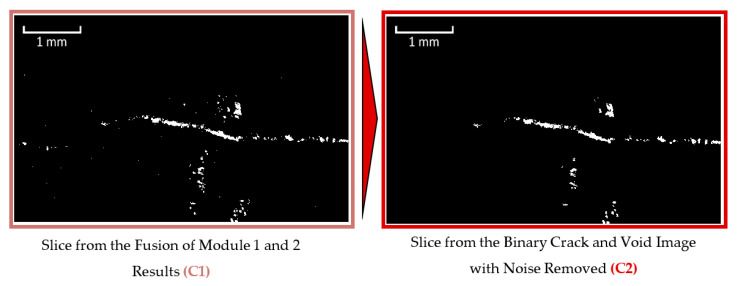
Removal of noise from binary crack and void image (based on figure in [10]).

**Figure 12 materials-13-03921-f012:**
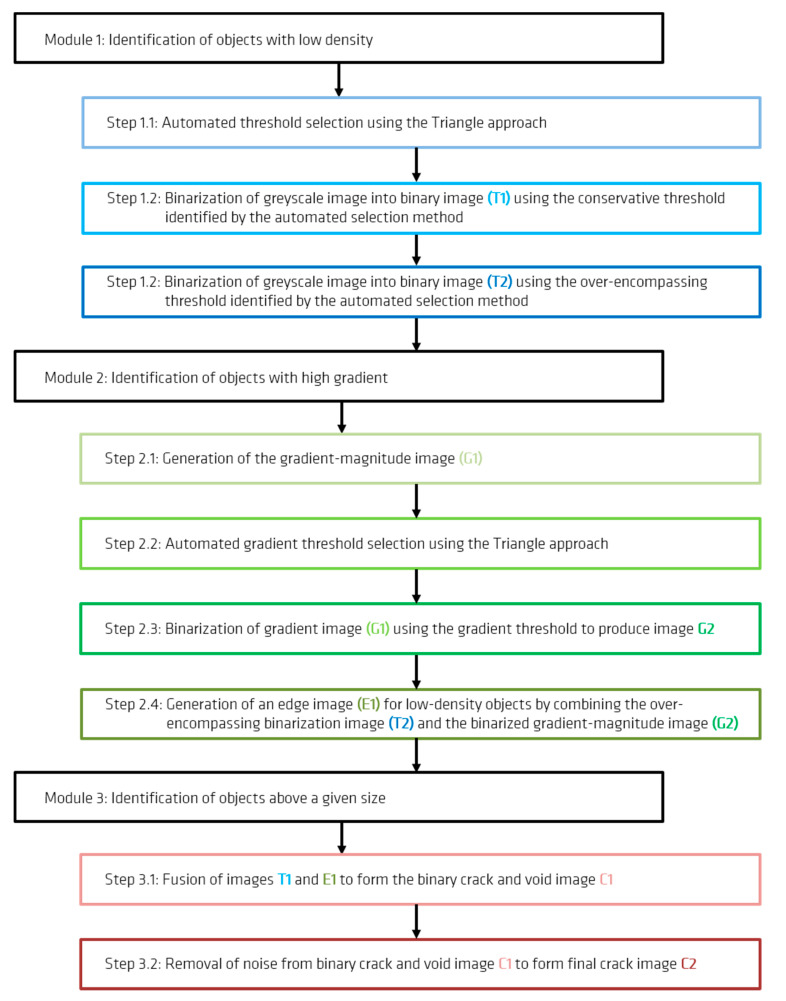
Flowchart of crack-detection process (based on figure in [10]).

**Figure 13 materials-13-03921-f013:**
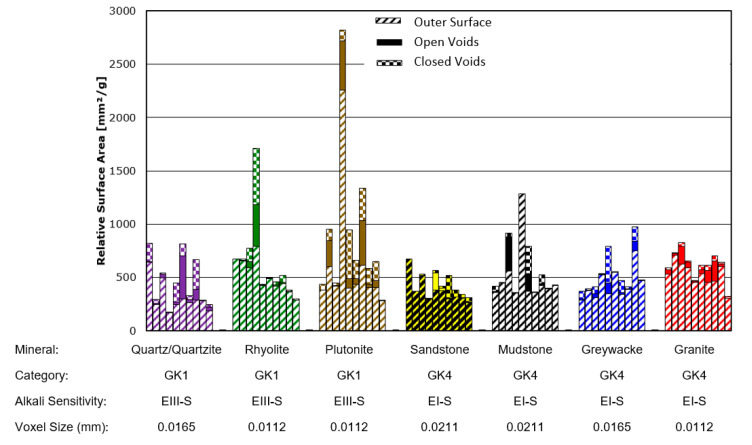
Measured surface area for river-gravel type aggregates (based on figure in [10]).

**Figure 14 materials-13-03921-f014:**
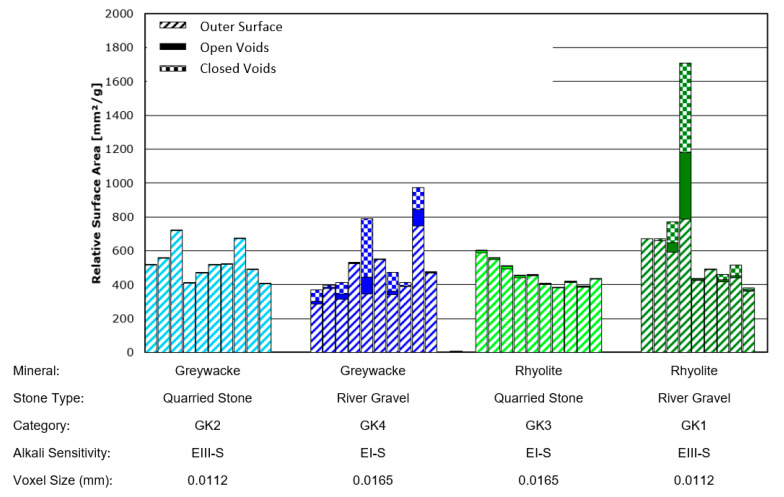
Surface area measurement comparison between quarried-stone (GK2 and GK3) and river-gravel (GK1 and GK4) aggregates of the same mineral types (based on figures in [10,11]).

**Figure 15 materials-13-03921-f015:**
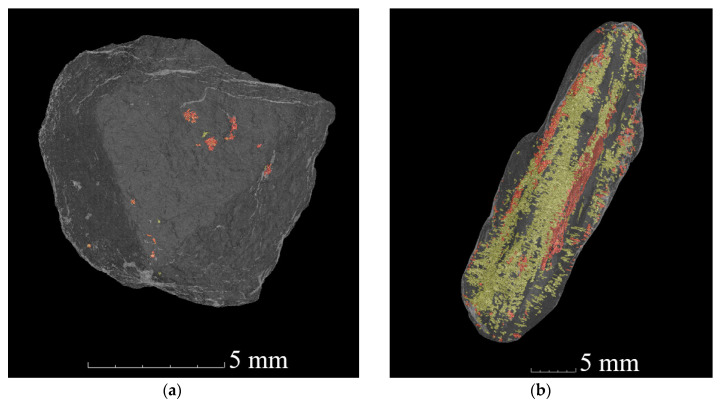
Example images of cracking within a quarried greywacke aggregate (**a**) and a river-gravel type greywacke aggregate (**b**). Red pixels denote externally accessible cracks/pores and yellow pixels denote externally inaccessible (closed) cracks/pores (reproduction of figures from [10,11]).

**Table 1 materials-13-03921-t001:** Categories of stone selected for cracking analysis.

Category	Stone Type	Alkali Sensitivity [40 °C-BV]
GK1	River Gravel	EIII-S
GK2	Quarried Stone (Greywacke)	EIII-S
GK3	Quarried Stone (Rhyolite)	EI-S
GK4	River Gravel	EI-S

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
