# Peer review of "Detection and Quantification of Cracking in Concrete Aggregate through Virtual Data Fusion of X-Ray Computed Tomography Images"

_materials, 2020, doi:10.3390/ma13183921_

Round 1

Reviewer 1 Report

This work is concerned the detection and analysis of microcracks in concrete aggregate with the use of X-ray computed tomography images method. The results analysis is preceded by an extensive review of the literature in the field of qualitative and quantitative micro-discontinuity analysis (cracks and pores) in CT and other investigations used of available software. In my opinion, this work is not greater scientific value, and the careful and correct analysis of the CT results is standard for identifying material defects in CT examinations. However, methodically this work is important, and may provide a guide for analyzing X-ray computed tomography images results.

Author Response

The authors would like to express their thanks for your review. In the revised version of the manuscript, the description of the methodology has been significantly improved in order to enhance its relevance and usefulness for materials researchers using X-ray computed tomography as a measurement method. These changes include a more thorough description of CT system operation and calibration as well as detailed individual measurement results included as a separate appendix. For a complete overview of all implemented changes, please see the revised, marked-up, version of the manuscript.

Reviewer 2 Report

The authors present a qualified NDT technique to identify cracks in gravel, and the paper seems to represent the same. Alkali-silica reactions have been studied extensively, however determining and mapping the exact root causes for crack development are an area of active investigation, so this paper does provide insights into this, Whilst the work is sound, there needs to be changes that will need to be incorporated before recommending this work to be published.

I have the following questions:

  1. Was the selection also based on mass of each of the gravel aggregate? If so, what influence did it have? Could you provide evidence perhaps in the form of a table.
  2. How was the gravel in the current form acquired? Are the cracks a result of the manufacturing process itself? Was any modification, in terms of artificial changes introduced into the gravel?
  3. Was the equipment calibration performed? If so, what was the voxel size to the linear dimension measure?
  4. For Fig 12, you have provided a measure of the area in-terms of number of voxels, how accurate was the measurement? Can you add an error bar?
  5. In Figure 12, one of the samples show a large size for Plutonite, was this because of a random selection? The size seems to be 3 times the average population size.
  6. For better interpretation, please tabulate the measurements from Fig 13, and provide a mean and std deviation to understand the variability in your measurement.
  7. What was the overall shape of the gravel, please add a digital image of the same.
  8. Mag-bars are missing in a lot of images, please add them to the images.
  9. Figure 13 - please organise as per the classification in Table 1. Also, are the measurements for the Rhyolite GK1 and Greywacke GK4 same to the ones in Figure 12?
  10. The references to all conference proceeding is not complete, the year of publication is missing. Please correct it.
  11. Further comments and suggestions can be found in the attached document.

Reviewer 3 Report

The research is relevant ; - I suggest to improve the introduction regarding the field of applications of X ray Computer Tomography in Medicine; I suggest to specify the number of sample used; Figure 8 legend must be improved ; Paper eell written, Conclusion consistent. Correct minor English error. Finally : ACCEPTED with MINOR REVISION

Round 2

Reviewer 2 Report

The revised manuscript reads well. The author has incorporated all the comments within the manuscript.